



# Evolution of the Arabian Sea upwelling in the past centuries and in the future as simulated by Earth System Models

Xing Yi, Birgit Hünicke and Eduardo Zorita

Helmholtz-Zentrum Geesthacht, Institute of Coastal Research, Max-Planck-Str.1, Geesthacht, 21502, Germany

*Correspondence to*: Xing Yi (email: xing.yi@hzg.de)

**Abstract.** Arabian Sea upwelling in the past has been generally studied based on the sediment records. We apply two earth system models and analyse the simulated water vertical velocity to investigate the variations of the coastal upwelling in the western Arabian Sea over the last millennium. In addition, two models with slightly different configurations are also employed to study the changes in upwelling in the 21st century under the strongest and the weakest greenhouse gas emission scenarios. With

a negative long-term trend caused by the orbital forcing of the models, the upwelling over the last millennium is found to be closely correlated with the sea surface temperature, the Indian summer Monsoon and sediment records. The future upwelling under the Representative Concentration Pathway (RCP) 8.5 scenario reveals a negative trend, in contrast with the positive trend displayed by the upwelling favourable along-shore winds. Therefore, it is likely that other factors, like water stratification in the upper ocean layers caused by the stronger surface warming overrides the effect from the upwelling favourable wind. No

significant trend is found for the upwelling under the RCP2.6 scenario, which is likely due to a compensation between the opposing effects of the increase in upwelling favourable winds and the stratification of the water column.

## 1 Introduction

Upwelling lifts the cold nutrient-rich water from deeper ocean layers to the surface, which cools the surface water and provides the biologically active layers with nutrients. It has a great impact on human activities. For instance, about 20% of the total marine

fish catches originate from upwelling regions that cover only 2% of the entire ocean area (Pauly and Christensen, 1995). Upwelling also affects the climate by cooling the sea surface temperature (SST) and by influencing the air-sea interactions associated with the variation of SST (Izumo et al., 2008). It has been suggested that, under the global warming scenario, coastal upwelling at global scale would be intensified as the upwelling favourable wind-stress would be strengthened due to the enhanced air-sea temperature gradient (Bakun, 1990). Numerous studies have provided insights into the upwelling variations

over the last few decades. Most of these studies focuses on the four major eastern boundary upwelling systems (EBUSs), namely, the California (Schwing and Mendelssohn, 1997), Canary (McGregor et al., 2007), Humboldt (Gutiérrez et al., 2011) and Benguela (Santos et al., 2012) upwelling systems. In support of the Bakun hypothesis, Narayan et al. (2010) found positive trends over the late 20th century in all these four coastal upwelling regions and Sydeman et al. (2014) also reported upwelling intensification in the major EBUSs after synthesizing the results from previous studies.


In addition to the major EBUSs, the Arabian Sea is one of the most productive regions in the world. The coastal upwelling in the western Arabian Sea is mainly driven by the southwest (SW) wind-stress which is induced by the Indian summer Monsoon (ISM). Since the link between the ISM and the upwelling is very pronounced, many studies have used the ISM as an indicator of the upwelling, and vice versa. However, direct observations of upwelling in terms of water mass vertical velocity over a long



time period are rare, thus, alternative upwelling proxies such as SST, surface wind-stress and sediment records have been generally applied. An alternative tool to analyse upwelling variability is provided by model simulations.

The knowledge of upwelling evolution in the past is crucial to understand and model the upwelling variations at present and to predict them in the future. A widespread approach to study upwelling in the past is through sediment records where the abundance of *G.bulloides* is stored. *G.bulloides* belongs to the phylum foraminifera and is very sensitive to upwelling variations so it is generally used as an upwelling proxy. Over the last millennium the Arabian Sea upwelling is reported to exhibit a slight decrease until approximately 1600 and an abrupt increase afterwards (Anderson et al., 2002; Feng and Hu, 2005; Sinha et al., 2011). As for the future, whereas a study focused on the Arabian Sea upwelling is not yet available, Wang et al. (2015) used the Coupled Model Intercomparison Project Phase 5 (CMIP5) (Taylor et al., 2012) model outputs to conduct an analysis on the 21st century upwelling under the future scenario of Representative Concentration Pathway (RCP) 8.5. They indicated that the upwelling intensity and duration are both increased at high latitudes in most of the major EBUSs except California. It is known that the external climate forcings are different over the last millennium and in the future. The responses to these forcings are also different among the major EBUS (Tim et al., 2016). Therefore, in the present study we analyse the impact of the external climate forcings on the Arabian Sea upwelling system and whether its response to the past can be linked with that to the future.

We investigate here the Arabian Sea upwelling variation over the last millennium by analysing model simulations of water vertical velocity and by comparing the model results with the observational sediment records to determine the existence of the long-term trends. The evolution of future upwelling in the Arabian Sea is also investigated by using models with slightly different configurations from the last millennium simulations. We compare the future upwelling under the RCP8.5 and RCP2.6 scenarios to gain the information on how the greenhouse gas emission level could affect the variation of upwelling.

## 2 Model and data

In this study, we select four earth system models from the CMIP5 model pool, which have the highest horizontal resolution to study the Arabian Sea upwelling. The outputs from two of these models are used for the last millennium study and the other two, which are similar to the previous ones but with slightly different configurations, are applied for the future scenarios. For the analysis over the last millennium, we use the paleo configuration of the Earth System Model of Max-Planck Institute for Meteorology (MPI-ESM-P) (Giorgetta et al., 2013) and the Last Millennium Ensemble (LME) project (Otto-Bliesner et al., 2016) of the Community Atmosphere Model Version 5 from CESM (CESM-CAM5) (Hurrell et al., 2013). To estimate the upwelling variabilities in the future, we apply the low resolution configuration of the MPI-ESM (MPI-ESM-LR) and the Community Climate System Model (CCSM4) which is the predecessor of the CESM. Note that the CCSM4 uses a different version of atmosphere model (CAM4) as the CESM (CAM5).

The MPI-ESM-P and the MPI-ESM-LR share the same horizontal resolutions of about 2 degrees for the atmosphere (192×96) and about 1 degree for the ocean (256×220). Their vertical resolutions are the same as well with 47 levels for the atmosphere and 40 levels for the ocean. In spite of this, the MPI-ESM-P is available for the last millennium (850-1849) and the MPI-ESM-LR covers the future period (2006-2100) simulated under the greenhouse gas emission scenarios. The CESM-CAM5 and the CCSM4 also have identical horizontal resolutions for the ocean which is about 1 degree on the longitude and 0.5 degree on the latitude (320×384). For the atmosphere, the CESM-CAM5 has a horizontal resolution of 2.5 degrees on the longitude and 1.875





degrees on the latitude (144×96), which is four times coarser than the CCSM4 (288×192). On the vertical, they share the same resolutions with 60 levels in the ocean and 30 levels in the atmosphere. The CESM-CAM5 covers the last millennium period (850-1849) and the CCSM4 covers the 21st century (2005-2100). The MPI-ESM-P and the MPI-ESM-LR each has an ensemble of three simulations that are forced by almost the same external forcings but started with different initial conditions, likewise, the

CESM-CAM5 and the CCSM4 have ensembles of many simulations so we have selected three simulations from each of them to compare with the MPI-ESM models. The external forcings used in the last millennium models are prescribed by a CMIP5 project defined standard protocol (Masson-Delmotte et al., 2013), which consists of solar irradiance variability (Vieira and Solanki, 2010), orbital forcing (Berger, 1978), volcanic activity (Gao et al., 2008; Crowley and Unterman, 2013), land-use changes (Pongratz et al., 2009) and greenhouse gases concentration (Flückiger et al., 2002; Hansen and Sato, 2004; MacFarling Meure et

al., 2006). For the future simulations only the anthropogenic forcing is changed based on the RCP scenarios.

The differences between simulations within each ensemble provide an estimation of the amplitude of the effect of internal climate variability, whereas the signal shared by all simulations will approximate the response to the imposed external forcing (Tim et al., 2016).

We investigate several variables that are modelled by the simulations, including upwelling velocity, sea surface temperature (SST), surface wind-stress, wind speed, and sea level pressure (SLP). The upwelling velocity is the "vertical velocity" in the CESM ensembles but has to be derived from the "vertical water mass transport" for the MPI-ESM ensembles. Since coastal upwelling in the western Arabian Sea occurs from 200 meters below the ocean surface (Brock and McClain, 1992), we average

the upper 200 meters of the data. We use the monthly data from the models and because the upwelling season starts in May and ends in September (Brock et al., 1991), only the summer months June, July and August (JJA) are selected. One exception is that for the SST we choose July, August and September (JAS) due to the lag in the response of SST to the upwelling (Rixen et al., 2000). In addition to the modelled data, we also apply the sediment records used by Anderson et al. (2002) to compare with our results.

**3 Arabian Sea upwelling in the last millennium**

The mean summer upwelling velocities in the Arabian Sea modelled by MPI-ESM-P and CESM-CAM5 for the last millennium are presented in Fig. 1. To avoid duplication, we only show the results of one simulation from each of these two simulation ensembles since the three simulations in each ensemble share very similar spatial patterns. The models present identical patterns of the summer upwelling in the Arabian Sea where strong upwelling occurs along the coast especially in the western Arabian

Sea, induced by the southwest wind-stress (Fig. 1a and 1b). Coastal upwelling in the western Arabian Sea is more intense in the MPI-ESM-P simulation, where the velocity can reach 1.5 m/day, than in CESM-CAM5 where the maximum velocity is around 1 m/day. The magnitude of these velocities is reasonable as it matches the estimated order of magnitude suggested from studies focused on present time period (Rixen et al., 2000; Shi et al., 2000). In addition, weak downwelling in the central Arabian Sea is found in both models. Downwelling here results from the convergence zone generated by the wind-stress curl (Thadathil et al.,

2008; Bauer et al., 1991). Thus, the spatial distribution of the vertical velocity is consistent with the Ekman pumping effect (Lee et al., 2000).



We average the upwelling along the coast and calculate the upwelling velocity time series of this area (Fig. 1c and 1d). This selected area spans about 5 degrees (500 km) from the coastline to the open ocean (shown within the green line in Fig. 1a and 1b). Although the coastal upwelling in this region is reported to extend about 100 km from the coast towards offshore (Rixen et al., 2000), such a narrow selection is not possible in our case due to the coarse resolutions of the models. In our selected study

area, the upward vertical velocity actually presents upwelling induced by the combination of the SW wind-stress and the wind-stress curl. Therefore, when using the simulated upwelling velocity in the study area, the effects of Ekman transport and Ekman suction are not separated. This is also reasonable because the along-shore SW wind-stress is highly correlated with the off-shore wind-stress curl since they are both driven by the land-sea pressure gradient. In general, the time series show that on average the mean upwelling velocity modelled by MPI-ESM-P is larger than the one simulated by CESM-CAM5 by around 0.3 m/day. The

multidecadal variation is, however, larger in CESM-CAM5. All six simulations by the two models reveal negative trends of the upwelling velocity. A detailed interpretation of these trends and their significance level will be presented in the Section 5.

Since upwelling lifts the cold water from the deeper layers to the ocean surface during the upwelling season, the SST is often anticorrelated with upwelling in the upwelling region. This relationship is well captured in both of the models (Fig. 2a and 2b).

Strong negative correlations are shown along the coastal upwelling regions and even greater ($r<-0.8$) values are found at some locations in the western Arabian Sea in the CESM-CAM5 model. We also notice the positive correlation areas in the central Arabian Sea where weak downwelling is revealed as shown in Fig. 1a and 1b. Downwelling is normally associated with convergence zones at the surface and weaker (stronger) downwelling is related to cooler (warmer) surface water in the same region. In terms of vertical velocity as it is in our case, downwelling is represented by the negative values of vertical velocities,

thus larger (smaller) velocities indicate weaker (stronger) downwelling. In principle, the SST should be negatively correlated with the vertical velocities in both upwelling and downwelling regions. However, Fig. 2a and 2b show positive correlations between SST and vertical velocity in the downwelling region of central Arabian Sea. This is likely due to the effect of horizontal advection (Turner et al., 2012; Lévy et al., 2007). For instance, during summer when stronger SW wind blows over the Arabian Sea, the coastal upwelling is enhanced. Such intensified upwelling raises more cold and deep water to strengthen the cooling of

SST along the coast. This cold water mass is then advected from the western to the central Arabian Sea via Ekman transport. In the central Arabian Sea, the downwelling is also enhanced as a result of the amplified Ekman pumping effect due to stronger wind-stress curl, but as mentioned before, in general the downwelling is weak so its influence on the local SST is very small. Therefore, the cooler water transported from the coast dominates the SST variability in the central Arabian Sea. Thus, the negative SST trend in the central Arabian Sea is associated with the drop in vertical velocity, which means the enhanced

downwelling is caused by the stronger wind-stress curl.

As a comparison with the sediment records which is considered as the observational data, Fig. 2c and 2d show the correlations between Arabian Sea upwelling and the *G.bulloides* abundance retrieved from the sediment cores RC2730 and RC2735 (Anderson et al., 2002). This record has been interpreted as an indicator of upwelling in this region (Peeters et al., 2002). We

only mark the location of RC2730 on the maps because these two cores are located very close to each other. The calculation of the correlation is performed on the 50-year averaged data during the overlapping years (1050-1849) of our modelled upwelling velocity and the *G.bulloides* abundance. To account for autocorrelation, we apply the nonparametric resampling method suggested by Ebisuzaki (1997) to estimate the significance level of the correlations. These filtered series presumably reflect the variations in the external forcing, or at least the externally forced component of the variability should be large. Since the external

forcing should be ideally the same in the observations and in the simulations, a positive correlation between both records should



be expected. However, the records from only two cores located at very close positions might not represent the upwelling in a broader area. In spite of this, the maps present significant correlations along the coast all the way to the northern Arabian Sea especially for the MPI-ESM-P model. Such correlations indicate that the models reasonably reproduce the variability of upwelling velocities at time scales that are presumably driven by the external climate forcing. The simulated records are thus comparable to the sediment records.

## 4 EOF analysis of upwelling

We perform an Empirical Orthogonal Function (EOF) analysis (von Storch and Zwiers, 2001) on the Arabian Sea upwelling to identify its main spatial variation patterns and the corresponding temporal evolutions. EOF analysis can identify the spatial patterns that describe the data variance by generating the EOF modes and their corresponding principal component (PC) time series, where each mode is ranked by its explained proportion of the total variance.

The leading two modes from the EOF analysis of the upwelling are given in Fig. 3. The ranks of the first two modes are switched between MPI-ESM-P and CESM-CAM5. The first mode from CESM-CAM5 (Fig. 3b) accounts for 41% of the total variance and reveals similar spatial patterns as the second mode of the MPI-ESM-P simulation (Fig. 3c) which accounts for 24% of the total variance. We find that these EOF modes are related to the interannual variations as they capture the spatial patterns of the mean state of the upwelling (Fig. 1a and 1b) where the different signs between coastal and central Arabian Seas show the contrast between these regions. Their PC time series (Fig. 4b and 4c) also have a strong positive correlation with the time series averaged from the coastal upwelling in the western Arabian Sea (Fig. 1c and 1d) for all the simulations respectively (Tab. 1).

Thus, the intensity of the coastal upwelling in the western Arabian Sea is in phase with the intensity of the upwelling in the rest coastal areas and also with the intensity of the downwelling in the central Arabian Sea. This result is supported by the correlation between upwelling and the Indian Monsoon Index (IMI). We calculate the IMI from the model derived U850 wind data based on the definition of Wang and Fan (1999). The IMI is significantly correlated to our upwelling time series in all the simulations (Tab. 1).

It is shown in Fig. 5a and 5b that the IMI is also correlated to the upwelling velocity in the rest coastal areas in the Arabian Sea and the negative correlation in the centre downwelling region indicates that the IMI contributes to the intensification of the downwelling as well. Since the IMI is calculated based on the wind field which is caused by the sea level pressure (SLP) gradient, we also apply the EOF analysis to the SLP field. We find that the time series of the coastal upwelling in the western Arabian Sea from both of the models are also significantly correlated with the PC time series of the second mode from the EOF analysis of the SLP (Tab. 1).

This EOF mode of SLP displays a clear spatial pattern representing the contrast between Africa and Indo-Asia (Fig. 5c and 5d). In general, the patterns revealed from the two models are very similar. However, the boundary line that separates the positive and negative EOF phases rotates anticlockwise in CESM-CAM5 comparing to that in MPI-ESM-P. This tilt might be responsible for the 8% more variance captured by the second EOF mode of SLP from CESM-CAM5 than that from MPI-ESM-P as well as the higher correlation between the SLP PC2 and the coastal upwelling time series derived from CESM-CAM5 than that from MPI-ESM-P (Tab. 1). An explanation might be that, with this tilt, the second EOF mode revealed from CESM-CAM5 captures a





contrast of the spatial patterns in the southern Arabian Sea, which contributes to the variance representing the SW wind-stress that highly correlates with the time series of the coastal upwelling in the western Arabian Sea.

## 5 Upwelling trends

In order to understand the millennial scale variability of the Arabian Sea upwelling, we calculate the long-term linear trends of the vertical velocity and determine their significance level using the Mann-Kendall test. Figure 6a and 6b show the upwelling trends derived from the two simulations. Both models reveal negative trends in the northern Arabian Sea and along the coast where the intense upwelling occurs. On the contrary, the central and eastern Arabian Seas display positive trends. Note that the central Arabian Sea is dominated by downwelling, and so the positive trends in this region actually indicate a weakening of downwelling, whereas in the eastern Arabian Sea the upwelling is slightly strengthened. However, in the western Arabian Sea, the region of our main focus, the upwelling velocity decreased over the last millennium, whereby the MPI-ESM-P model displays a more significant reduction than the CESM-CAM5 model. The upwelling velocity trends share very similar spatial patterns with the first mode of MPI-ESM-P (Fig. 3a) and the second mode of CESM-CAM5 (Fig. 3d) from the EOF analysis. In addition, their PC time series (Fig. 4a and 4d) also confirm the trends revealed in the upwelling velocity time series (Fig. 1c and 1d). The different signs in the EOF spatial patterns and the trends in the PC time series indicate the weakening of the western Arabian Sea coastal upwelling. The trends revealed from the time series are small in terms of their amplitudes over a thousand years compared to the mean upwelling velocity. However, negative trends of western Arabian Sea upwelling are shown consistently in all the six simulations by the two models despite of the different internal conditions used for the simulations. Thus, the overall picture of these trends is highly statistically significant. Here we use a classical sign test (von Storch and Zwiers, 2001) to determine the overall significance level of the trends. The sign test is usually applied in statistical climatology, for instance, Tebaldi et al. (2011) employed this method to test the significance level of the precipitation trends simulated in an ensemble of climate simulations. Similar to their work, in our case the null hypothesis is that there is no trend in the simulated upwelling in the last millennium. If we use an unbiased estimator to estimate the trends in the simulations, the number of simulations n out of N yielding negative trends follows a binomial distribution (a flip coin distribution) and the level of significance can be estimated as the number n in this distribution that yields the desired level of statistical significance. That is, if we consider only the sign of the trends, given that all the six simulations are independent samples, the probability that all of them show negative trends by chance is $0.5 \times 6 = 0.01$. So the overall trend analysis is statistically significant at p = 0.01, not to mention that the trends are not only all negative but some are also statistically significant at the individual p = 0.05 level, which means the overall significance is much stronger than p=0.01. In addition, the ensemble means of the two models also reveal significant negative trends at the p=0.01 level (not shown). Therefore, the weakening of upwelling in this region is very likely a robust feature in the simulations.

These trends are likely induced by the external forcing used to drive the models. Among all the external forcings, only the orbital forcing displays the long-term millennial scale trend, so that the identified upwelling trends can presumably be attributed to the orbital forcing. The change in orbital forcing is a significant cause of the SST trends revealed in the tropical and subtropical regions (Lorenz et al., 2006). The monsoon was strengthened during the Mid-Holocene as a result of the enhanced SLP gradient between land and ocean due to the increased land-sea thermal contrast induced by the orbital forcing (Jiang et al., 2012). More recently, Gill et al. (2017) have shown negative trends of the Arabian Sea upwelling and the Indian Monsoon from past to present, which is also linked to the orbital forcing.



To confirm the effect of the orbital forcing, we calculate the trends of the SLP (Fig. 6c and 6d) and the SW wind-stress (Fig. 6e and 6f) since the upwelling is mainly forced by the SW wind-stress which is generated from the SLP gradient. The spatial patterns of the SLP trends show clearly that in mid latitudes the SLP tends to increase and in low latitudes the SLP tends to

decrease. These patterns are quite likely resulting from the effect of orbital forcing, as similar long-term changes are also derived from the differences between equilibrium mid-Holocene and present climate simulations (Braconnot et al., 2002). Moreover, we have also compared our results with those from the individual forcing experiments (orbital-forcing-only and solar-forcing-only) of the CESM-CAM5 model. It is found that both upwelling and SLP in the orbital-forcing-only simulations present similar trend patterns as what is shown in Fig. 6 but they are not revealed from the solar-forcing-only simulations. This further ascertains the

impact of the orbital forcing. With positive trends in the mid latitudes and negative trends in the low latitudes, the SLP contrast between these areas is reduced, which further affects the wind in the Arabian Sea. During the summer upwelling season, this SLP contrast drives the SW wind so when the SLP contrast is reduced the SW wind is also weakened. Figure 6e and 6f present the negative trends of SW wind-stress as expected. In general, the trend from MPI-ESM-P is larger than that from CESM-CAM5. Since the western Arabian Sea coastal upwelling is significantly linked to the SW wind-stress in the Arabian Sea, the

negative trends of SW wind-stress can cause the decrease in the upwelling velocity (Fig. 6a and 6b). A larger trend of SW wind-stress in the case of the MPI-ESM-P model than in the CESM-CAM5 model also results in stronger trend of upwelling. Therefore, the weakening of the coastal upwelling in the western Arabian Sea is induced by the reduction of the SW wind-stress which results from the long-term change of the SLP contrast between mid and low latitudes due to the orbital forcing.

## 6 Future scenarios

In addition to the analyses of the last millennium simulations, we study the trends in the future scenarios as well. Figure 7 presents the time series of the SST, the SW wind-stress and the upwelling velocity modelled by MPI-ESM-LR and CCSM4 under the scenario RCP8.5. In order to be consistent with the last millennium study, these time series are calculated from the same selected area shown in Fig. 1.

All the simulations by the two models are quite consistent regarding the results for the RCP8.5 scenario by showing positive trends in SST and SW wind-stress but negative trends in upwelling velocity. The SST (Fig. 7a and 7b) increases significantly under the effect of the greenhouse gas emission. In the MPI-ESM-LR model (Fig. 7a) the SST begins with lower values than that in the CCSM4 model (Fig. 7b) but they rise to similar values by the end of the simulations so the SST modelled by the MPI-ESM-LR has a larger trend than the CCSM4. The SW wind-stress (Fig. 7c and 7d) is also strengthened due to the enhanced

contrast between the surface heating over the land and the sea. This finding of enhanced upwelling favourable winds is consistent with the result of Menon et al. (2013). They presented that the simulated changes in the Monsoon circulation are quite robust across the CMIP model ensemble and consistent with the changes simulated by the MPI-ESM and CESM models, including the strengthening of the upwelling favourable winds in the future under RCP8.5 scenario. Although all the simulations by both models show positive trends, the CCSM4 simulations (Fig. 7d) reveal more distinct and significant trends than the MPI-ESM-LR

(Fig. 7c). As a result of the intensification of the wind-stress the upwelling velocity is expected to increase as well. However, negative trends of upwelling velocity are found in all the simulations (Fig. 7e and 7f) especially for the MPI-ESM-LR (Fig. 7e). In addition, we have also examined the simulated upwelling in the CESM model under the RCP8.5 scenario in the 21st century and it shows similar results as in the CCSM4 model with significant trends at the same order, despite the differences between



two models. Therefore, an interesting question arises as to what is causing the negative trend in upwelling velocity given that the upwelling favourable wind-stress is projected to increase. Figure 7 indicates that with stronger trends in SST and weaker trends in SW wind-stress, the MPI-ESM-LR simulations present a stronger decrease in upwelling whereas the CCSM4 simulated upwelling decreases less as the SST reveals weaker trends and the SW wind-stress shows stronger trends in CCSM4. Thus, it is

likely that the increase in surface water stratification caused by warming of the upper ocean layers in the RCP8.5 scenario could override the effect of the upwelling favourable wind. The upwelling weakens more when the warming is stronger or the wind is less intensified, and vice versa.

This hypothesis is confirmed by the results simulated under the RCP2.6 scenario where both models show weaker warming of

surface water than the RCP8.5 simulations but no consistent trend in the SW wind-stress or the upwelling. That is, without the strong increase in SST the water stratification in the upper layers is less intense so the decrease in upwelling is weaker and is compromised by the effect of the SW wind-stress. Note that the correlations between the upwelling and the SW wind-stress are similar in RCP8.5 and RCP2.6 scenarios (Tab. 2) so the influences of the wind on the upwelling are at the same level.

In addition, Fig. 8 presents the trends of upwelling velocity and water temperature at all levels in the upper 200m, which further supports that the upwelling reduction under RCP8.5 scenario is caused by the surface water stratification due to the strong warming. It is very clear, especially in the CCSM4 plots (Fig. 8c and 8d), that in the upper 50 m to 100 m the upwelling velocity show larger negative trends when the trends of the water temperature increase. At around 50 m depth the temperature trends reach their peaks and the upwelling trends also get to the lowest values (in the surface layer above 100 m). The upwelling is

actually most intense at this layer but it is most reduced as well, which causes the drop in the upwelling time series shown in Fig. 7. Therefore, in spite of the positive correlation between the SW wind-stress and the upwelling, the warming of the upper ocean layers under the RCP8.5 scenario reduces the upwelling velocity more efficiently.

## 7 Conclusion and Discussion

The variability of the Arabian Sea upwelling over the last millennium and in the 21st century are investigated in this study. We

statistically analyse and compare the outputs of four earth system model simulation ensembles and conclude:

- Over the last millennium, the modelled upwelling velocity has a strong correlation with the SST and the IMI as well as the observed *G.bulloides* abundance. This indicates a close connection among the Arabian Sea upwelling, the Indian summer Monsoon and the sediment records.


- A negative long-term trend is found in the upwelling velocity over the last millennium, which is resulting from the orbital forcing of the models. In the last 300 years, however, the upwelling reveals a positive trend, which matches the observations in the sediment records.

- The correlation between upwelling and the SW wind-stress remains at the same level under the RCP8.5 and RCP2.6 scenarios in the future but the stronger greenhouse gas emission scenario leads to opposite trends in the SW wind-stress and the upwelling velocity. The positive trend in the SW wind-stress is caused by the intensified SLP gradient between land and ocean due to the warming effect. The upwelling velocity shows a negative trend because under the stronger



scenario, the warming of the sea water tends to stabilize the surface layer, which overrides the effect of the SW wind-stress and interrupts the upwelling.

One limitation of this study is that it relies to a great extent on the realism of the models employed, which makes it very difficult to validate the results as there are no available direct observations on upwelling over the time span. However, we do find a connection between our modelled upwelling velocity and the observational sediment records. They present a significant correlation (Fig. 2) which indicates the effect of the external forcing. In addition, they also have similar patterns in the time series in terms of the trends. It can be noticed that the long-term trend revealed from the upwelling time series (Fig. 1) tends to flip, especially in the CESM-CAM5 model, at around the year 1550. This flip was reported by Anderson et al. (2002) where they

reconstructed the upwelling index (monsoon winds) from the sediment records over the last millennium. They found a slightly negative trend from 1000 to 1500 and a positive trend thereafter, which is very similar to our finding. Thus, we divide our upwelling time series into two periods based on the flip and calculate the trends separately (Tab. 3).

It is very clear that five out of six of the simulations indicate a flip in the long-term trend at 1550 as the signs of the trends

change from negative to positive after this point (except for the CESM-CAM5 r1), which is at about the same time as the *G.bulloides* records. To explore the flip in the modelled upwelling time series, we select a range of years from 1550 to 1650 when the flip might happen. We consider each of the years within this range as a flip point and calculate the trends before and after this point, that is, the trend from 850 to the selected year and the trend from the selected year to 1850. We apply this to all the six simulation of the MPI-ESM-P and the CESM-CAM5 models and combine the results. The histograms in Fig. 9 show that

95% of the trends before the flip point are negative and 73% of the trends after the flip point are positive. Among all the trends before and after the flip point, the percentage of the significant ($p$-value$<0.1$) ones are 35% and 25%, respectively. More importantly, the significant trends are consistent on the signs in both cases (negative before flip and positive after flip). Therefore, our models produce realistic upwelling time series that agree with the observational sediment records. However, uncertainties remain as the exact time when the trends change signs varies from simulation to simulation, which might be caused

by the different initial conditions applied for each simulation.

Model resolution also has an impact on the realism of the results (Small et al., 2015; Ranjha et al., 2016). However, it is still unclear what resolution is most suitable to model the Arabian Sea coastal upwelling system. In our study, the MPI-ESM uses a coarser horizontal resolution (256×220) than the CESM (320×384) and produces the upwelling velocity with larger means but

less variability, as well as larger trends. The upwelling velocity simulated by the CESM, however, has stronger correlation with the wind and the wind based IMI. Whether these differences are caused by the different resolutions is uncertain but this comparison might shed light on how the modelled Arabian Sea upwelling responds to the change in the model resolutions.

A recent study found a reduction in the future upwelling at low latitudes in most of the four EBUSs (Wang et al., 2015). In our

analysis for the future scenario RCP8.5, the upwelling velocity also reveals a negative trend whereas the SW wind-stress presents a positive trend. Thus, the reduction of the upwelling is not caused by the wind-stress, although they are still significantly correlated. The analysis for the RCP2.6 scenario does not show a trend in either the upwelling or the SW wind-stress but it shows strong correlation between upwelling and the wind-stress as well as the RCP8.5 scenario. Wang et al. (2015) suggested several mechanisms that could lead to a reduction in the upwelling at low latitudes such as strengthened downwelling favourable winds,

weakened upwelling favourable winds, and enhanced water stratification due to the greenhouse warming. In our case, it is the




stratified water under the RCP8.5 scenario that reduces the upwelling but this is not shown in the RCP2.6 scenario. Therefore, it remains for further study to investigate at what level of future emissions between the strongest and the weakest warming scenarios the water stratification and the effect of wind-stress approximately balance.

**Acknowledgements**

This work is funded by the Cluster of Excellence Integrated Climate System Analysis and Prediction (CliSAP) Project B3. We thank the Max-Planck-Institute for Meteorology for providing the MPI-ESM model data. All the other publicly available data used in this study are gratefully acknowledged.

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

|  | MPI-ESM-P upwelling TS | | | CESM-CAM5 upwelling TS | | |
|---|---|---|---|---|---|---|
|  | r1 | r2 | r3 | r1 | r2 | r3 |
| IMI | 0.64 | 0.61 | 0.60 | 0.64 | 0.66 | 0.65 |
| Upwelling PCs* | 0.79 | 0.85 | 0.83 | 0.92 | 0.93 | 0.93 |
| SLP PC2 | 0.61 | 0.50 | 0.55 | 0.84 | 0.87 | 0.83 |

(*) Upwelling PC2 is used for correlations within the MPI-ESM-P simulations but upwelling PC1 is used for CESM-CAM5

10 **Table 2: Correlation coefficient between the upwelling velocity and the SW wind-stress modelled by MPI-ESM-LR and CCSM4 for the RCP8.5 and RCP2.6 scenarios. All the correlation coefficients are calculated from the detrended time series and are significant at the 95% significance level.**

|  | MPI-ESM-LR | | | CCSM4 | | |
|---|---|---|---|---|---|---|
|  | r1 | r2 | r3 | r1 | r2 | r3 |
| RCP8.5 | 0.5457 | 0.4787 | 0.5420 | 0.4844 | 0.5500 | 0.5321 |
| RCP2.6 | 0.4270 | 0.4259 | 0.4752 | 0.7569 | 0.6391 | 0.6879 |

15 **Table 3: Upwelling trends (m/day per 1000 years) calculated from the two models for different time periods: the entire time period (850-1849), the first 700 years (850-1549), and the last 300 years (1550-1849).**

|  | MPI-ESM-P upwelling trend | | | CESM-CAM5 upwelling trend | | |
|---|---|---|---|---|---|---|
|  | r1 | r2 | r3 | r1 | r2 | r3 |
| 850 - 1849 | -0.0187* | -0.0062 | -0.0164* | -0.0013 | -0.0129 | -0.0138 |
| 850 - 1549 | -0.0120 | -0.0052 | -0.0105 | -0.0005 | -0.0028 | -0.0343* |
| 1550 - 1849 | 0.0567 | 0.0862* | 0.0062 | -0.0100 | 0.1484* | 0.0905 |

(*) Trends that are significant at the 95% level



**Figures**

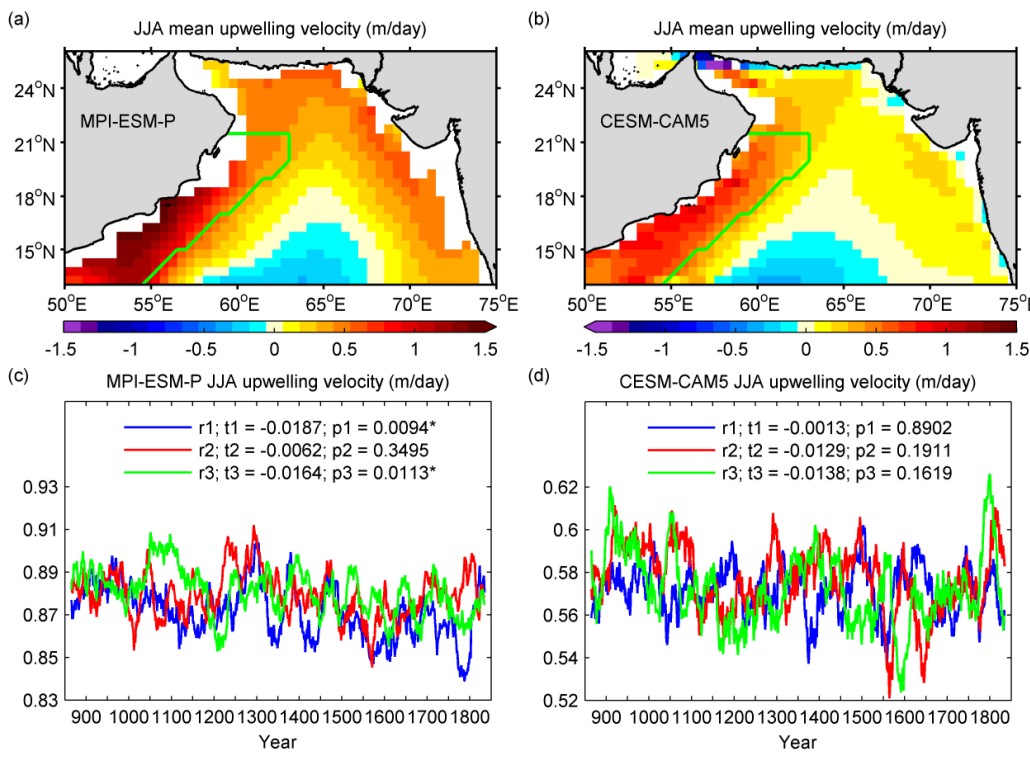

Figure 1: Spatial patterns of JJA mean Arabian Sea upwelling velocity modelled by (a) MPI-ESM-P and (b) CESM-CAM5. Time series of JJA coastal upwelling velocity in the western Arabian Sea modelled by (c) MPI-ESM-P and (d) CESM-CAM5. The area selected for calculating the variable time series is shown in (a) and (b) within the green line. The plotted time series are smoothed by a 31-year moving mean window and r1, r2, r3 represent the three simulations. t1, t2, t3 are the trends (per 1000 years) of each simulation with p1, p2 and p3 indicating their p-values respectively and the star symbols mark the simulations that pass the significance test at the 95% level.



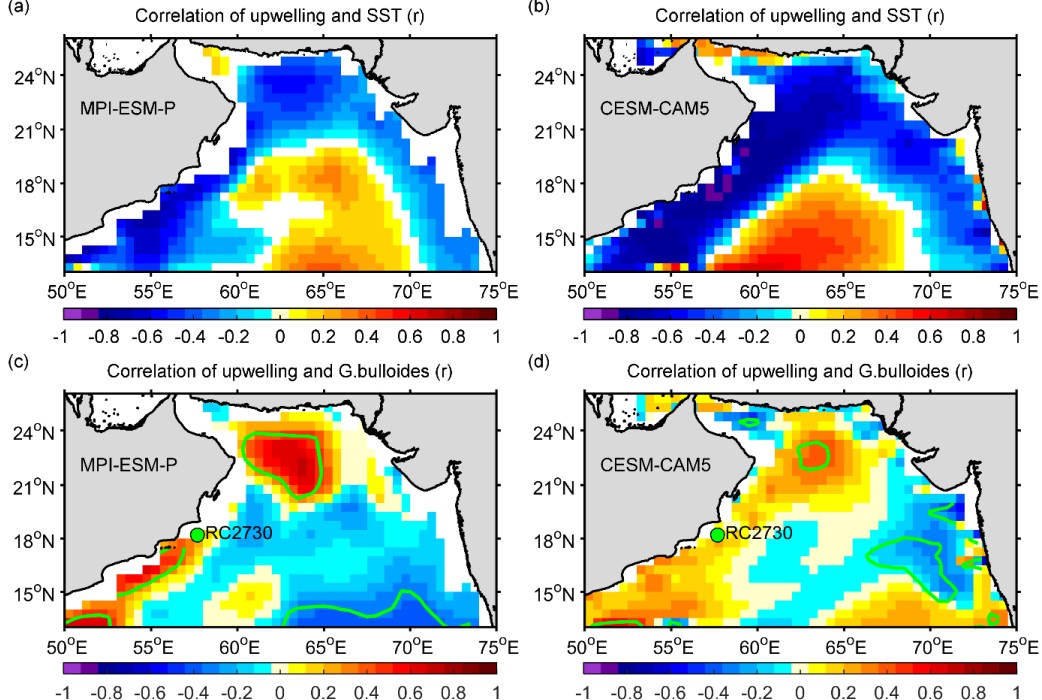

**Figure 2: Correlation coefficient between the SST and the upwelling velocity modelled by (a) MPI-ESM-P and (b) CESM-CAM5 with significance level greater than 95% (p-value<0.05). Correlation coefficient between the G.bulloides abundance and the upwelling velocity simulated by (c) MPI-ESM-P and (d) CESM-CAM5. The green point marks the location of the sediment core where the G.bulloides abundance is recorded. The regions surrounded by the green contour lines present significant correlations which are at a higher level than 90% (p-value<0.1). All the correlation coefficients are calculated from the detrended time series.**





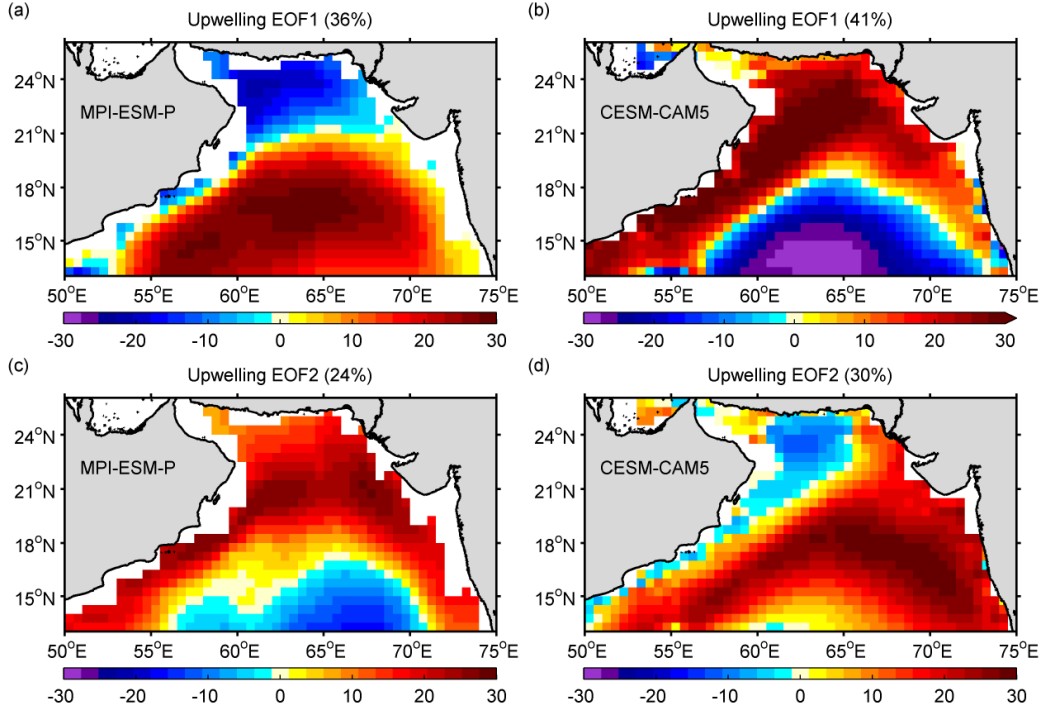

**Figure 3: First EOF mode of JJA Arabian Sea upwelling velocity modelled by (a) MPI-ESM-P and (b) CESM-CAM5 and second EOF mode from (c) MPI-ESM-P and (d) CESM-CAM5 with their explained variance in brackets.**





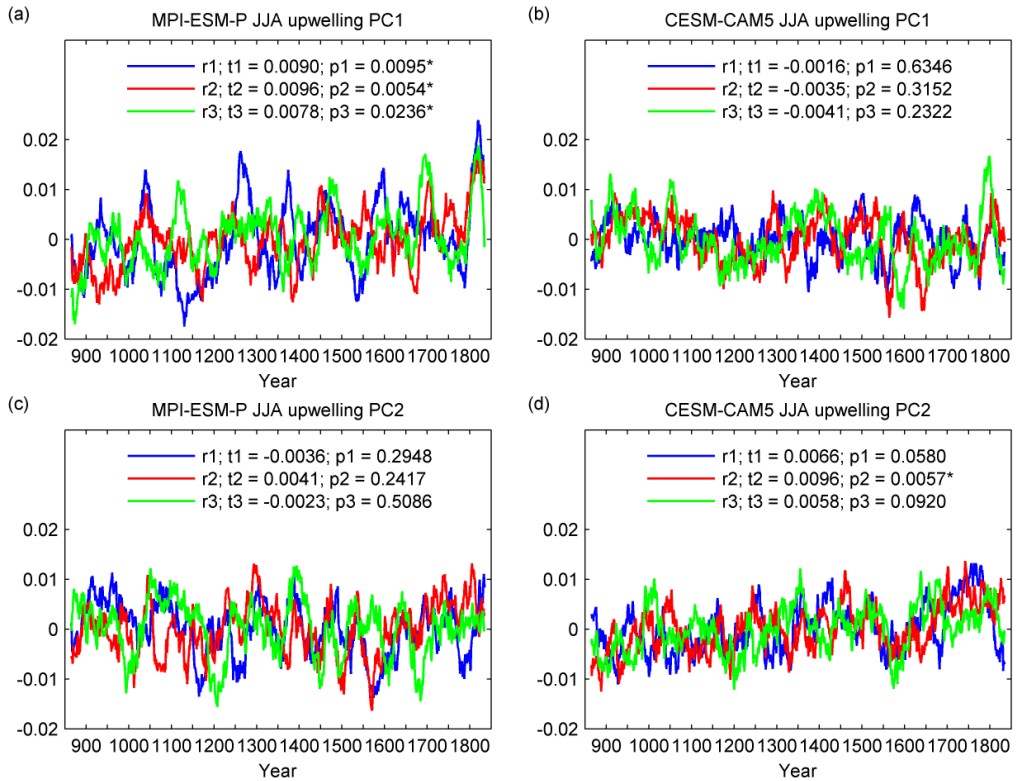

**Figure 4: First principal component (PC) time series corresponding to the first EOF mode for all the simulation in (a) MPI-ESM-P and (b) CESM-CAM5 and second PC time series of (c) MPI-ESM-P and (d) CESM-CAM5. The plotted PCs are smoothed by a 31-year moving mean window and r1, r2, r3 represent the three simulations. t1, t2, t3 are the trends (per 1000 years) of each simulation with p1, p2 and p3 indicating their p-values respectively and the star symbols mark the simulations that pass the significance test at the 95% level.**



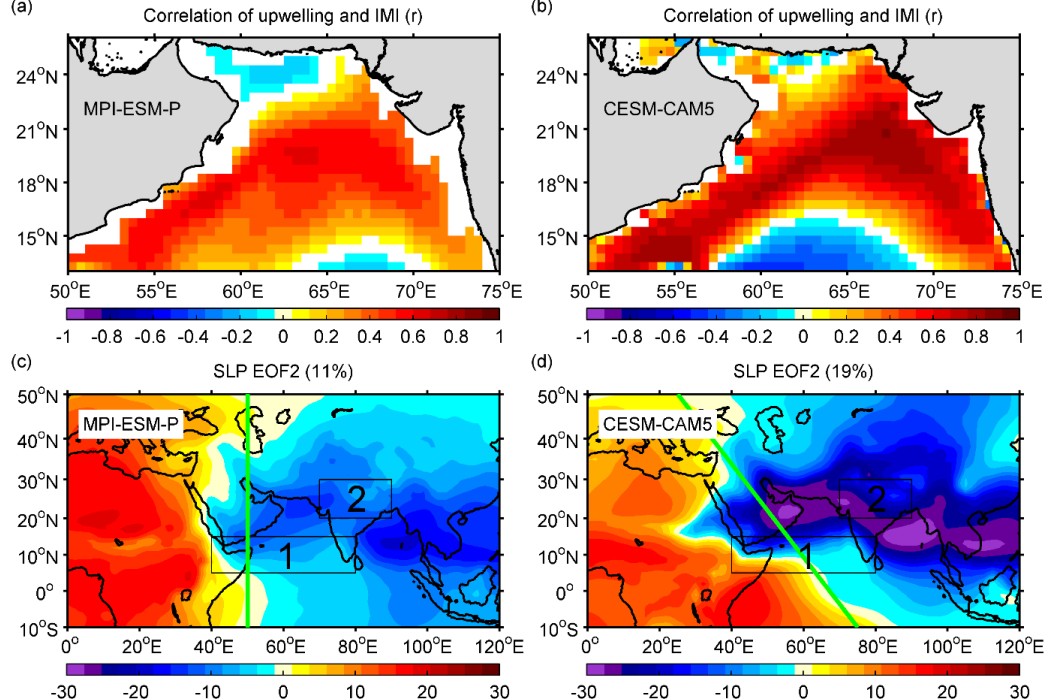

**Figure 5: Correlation coefficient between the upwelling velocity and the IMI modelled by (a) MPI-ESM-P and (b) CESM-CAM5. The IMI is calculated by subtracting the U850 wind averaged in box2 from that in box1 (boxes shown in c, d). All the correlation coefficients are calculated from the detrended time series and are significant at the 95% significance level. The second EOF mode of JJA SLP modelled by (c) MPI-ESM-P and (d) CESM-CAM5 with their explained variance in brackets. The green lines demonstrate approximately the simplified boundaries that divide the positive and negative EOF phases.**





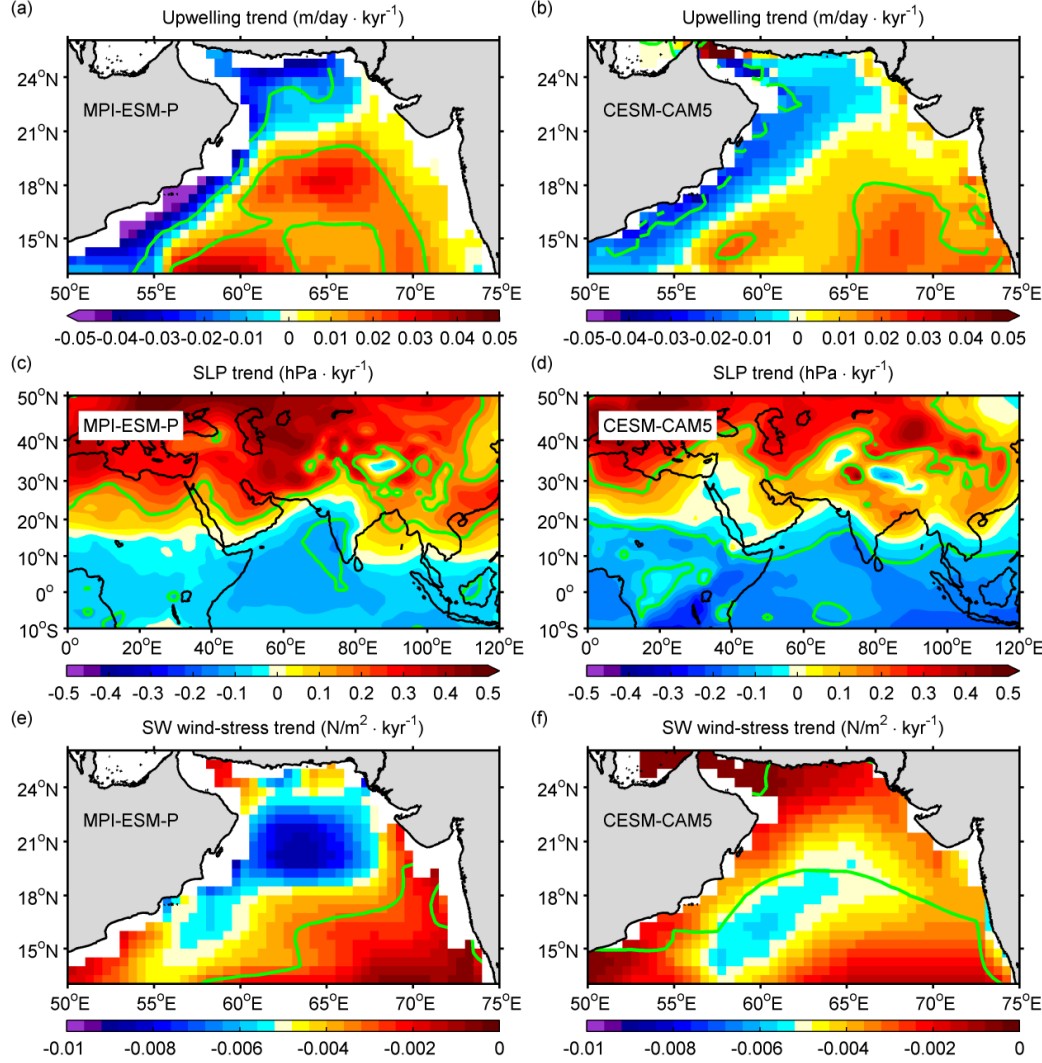

**Figure 6: Long-term trends of JJA upwelling velocity modelled by (a) MPI-ESM-P and (b) CESM-CAM5. Long-term trends of JJA SLP modelled by (c) MPI-ESM-P and (d) CESM-CAM5. Long-term trends of JJA SW wind-stress in the Arabian Sea modelled by (e) MPI-ESM-P and (f) CESM-CAM5. The regions surrounded by the green contour lines present significant trends which are at a higher level than 90% (p-value<0.1).**







**Figure 7: Time series of (a) SST, (c) SW wind-stress and (e) upwelling velocity modelled by MPI-ESM-LR under the scenario of RCP8.5 for the 21st century. (b), (d) and (f) are the same variables respectively modelled by CCSM4. r1, r2, r3 represent the three simulations of each model. t1, t2, t3 are the trends (per 100 years) of each simulation with p1, p2 and p3 indicating their p-values respectively and the star symbols mark the simulations that pass the significance test at the 95% level.**





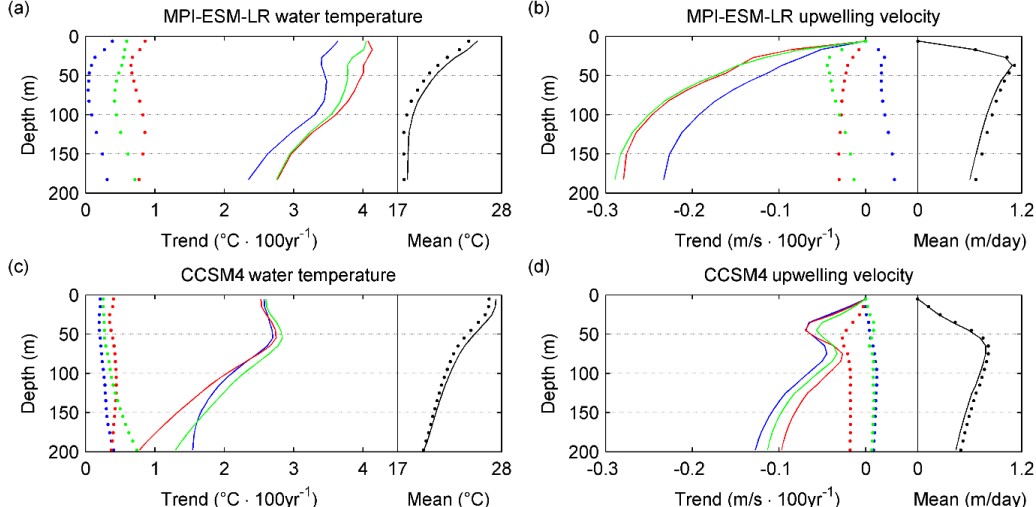

**Figure 8: Vertical trends of (a) water temperature and (b) upwelling velocity in the upper 200m modelled by MPI-ESM-LR. (c) and (d) are the same but by CCSM4. Solid lines are the results for the RCP8.5 scenario and dotted lines are for the RCP2.6 scenario. The coloured lines show the trends of r1 (blue), r2 (red), and r3 (green) in each model. The black lines on the right of each panel present the mean value of the current variable.**



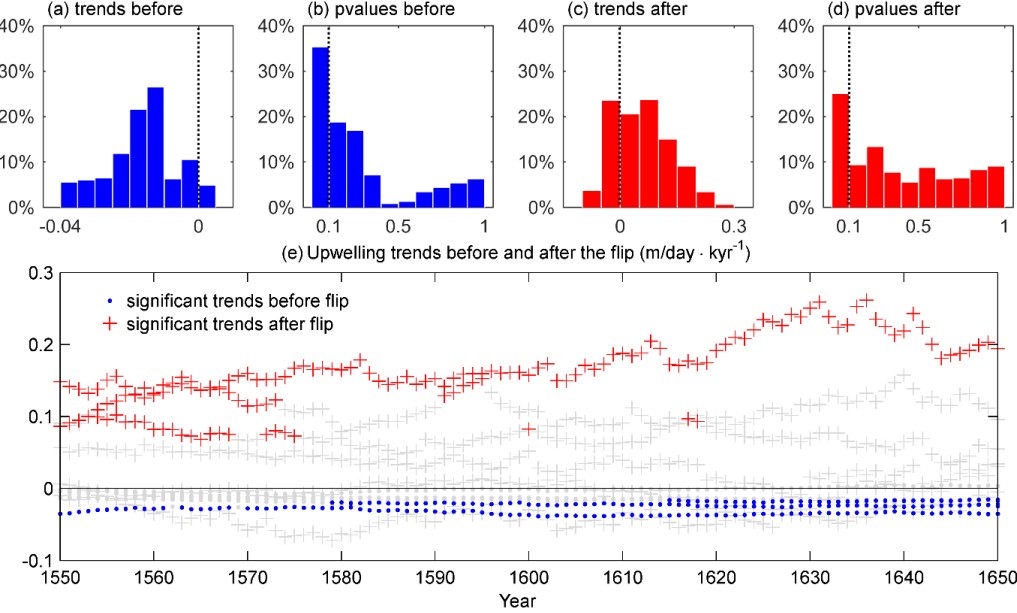

Figure 9: Histograms of (a) trends and (b) p-values for the period before the selected flip point. Histograms of (c) trends and (d) p-values for the period after the selected flip point. (e) Trends before (dot) and after (plus) each flip point from 1550 to 1650. The colorred markers show significant trends with p-value < 0.1.

