# Peer review of "Evolution of the Arabian Sea upwelling in the past centuries and in the future as simulated by Earth System Models"

_Climate of the Past, 2018_

## Referee Comment (RC1) · Anonymous Referee #1 · 8 Jul 2019

This study analyzed the simulated water vertical velocity to investigate the variations of the coastal upwelling in the western Arabian Sea over the last millennium. The two models are also employed to study the changes in upwelling in the 21st century under the strongest and the weakest greenhouse gas emission scenarios. With a negative long-term trend caused by the orbital forcing of the models, the upwelling over the last millennium is found to be closely correlated with the sea surface temperature, the Indian summer Monsoon and sediment records. The future upwelling under the Representative Concentration Pathway (RCP) 8.5 scenario reveals a negative trend, in contrast with the positive trend displayed by the upwelling favorable along-shore winds. Therefore, it is likely that other factors, like water stratification in the upper ocean layers

caused by the stronger surface warming overrides the effect from the upwelling favorable wind. The paper is overall well written and contributes to improve our knowledge of how the upwelling evolves from the past to future. Nevertheless, I consider that some important points must be addressed before the paper can be published. General comments: 1. Orbital forcing is not the major forcing of climate variability during the last millennium, except for the millennium-scale decreasing trend, is there any relationship between the upwelling and the volcano activities? 2. Since the link between the ISM and the upwelling is pronounced, how is the relationship between ENSO and the upwelling? 3. The authors showed the CESM ensemble results, have you evaluated the ensemble spread to measure significance or uncertainties, for example including ensemble ranges would be useful. 4. The future upwelling under RCP8.5 scenario reveals a negative trend, given that the upwelling favorable wind-stress is projected to increase. This is opposite from our common understanding that coastal upwelling at global scale would be intensified as the upwelling favorable wind-stress would be strengthened due to the enhanced air-sea temperature gradient under the global warming scenario. The authors explain this from the point of surface water stratification, is there a critical time when we could explain by the two different theories?

---

## Referee Comment (RC2) · Anonymous Referee #2 · 9 Jul 2019

The present study of Yi et al. uses earth system model experiments to study the evolution of Arabian Sea upwelling in the context of a global warming scenario. First, they use vertical water transport velocity as a parameter to describe upwelling intensity and compare the results over the last millennium to the atmospheric Indian Monsoon Index and the sedimentary record of G. bulloides abundances. This information is then used to run a different set of models under Representative Concentration Pathway (RCP) 8.5 scenario, that finds a decrease of upwelling intensities in contrast to increasing along-shore wind intensities. The topic is potentially interesting for economy and scientific research, but the approach raises several major concerns, preventing me from recommending publication.

[Figure]

1.) I would be very cautious about the choice of the models resolutions to be appropriate to investigate coastal upwelling in the Arabian Sea. The spatial extent (e.g., Figs. 1 a and b) shows that coastal areas are largely blank, especially for MPI-ESM, but also for CESM-CAM5. It is clear from previous studies that coastal upwelling is restricted to the vicinity of the coast, approx. 90 km offshore (Rixen et al., 2000). In fact, the authors mention this issue on p.4 l.3 but miss to discuss what implications this might have on the reliability of the results. It was further discussed by Praveen et al. (2016 Geophys.Res.Lett.), that changes in upwelling under future warming scenarios are regionally limited and spatially inhomogeneous, finding that 1x1 degrees atmospheric resolution is not adequate for studying the coastal current system. The study of Yi et al. uses models with more than four times coarser resolutions and it has to be discussed if a potential increase of coastal vertical water transport might not be obscured by the coarse resolution.

2.) Although the approach of testing the models against observational data seems to be of minor importance in the study, I highly question the approach used here with G. bulloides data. My first concern is, that it is not clear from Fig. 2c) if cores RC2730 and RC2735 are actually within the modelled grid cell or outside. If the stations are covered, they are at the very edge of the models coverage anyway. Secondly, the actual correlation of the two time series, Oman margin upwelling intensity and G. bulloides abundance, is not given in Table 1, where I would expect it. It seems from the colour coding in Fig. 2 c) and d) that it is in the range of r=0.2 to 0.4. However, both models show a significant positive correlation only in the northern part of the basin, an area of open-ocean upwelling and outside the area that was used for calculating the time series. This is not explained in the text and I suppose that it is not appropriate to verify the model results in this manner. I would also expect to see the G. bulloides data parallel to the upwelling time series data, as this would give a more obvious connection of the relationships. Especially as the authors discuss a "flip" at 1550 also be evident in the G. bulloides record, it has to be illustrated in a clearer way.

3.) The finding of increased upwelling favourable winds under RCP8.5 scenario together with a negative trend in upwelling intensity is explained with a likely overriding effect due to increased surface water stratification and warming SST. However, I miss a discussion of the results in context to previous studies of modelled upwelling in the Arabian Sea. Although using historical simulations over a much shorter period of time, Roxy et al. (2016 Geophys. Res. Lett.) for example found decreasing trends of phytoplankton productivity by using a similar set of simulations and also inferred an overriding effect of near-surface stratification as the main cause.

---

## Author Comment (AC1) · 12 Aug 2019

We thank the reviewer for detailed reading of the manuscript and for the suggestions for improvement. In the following, we quote the reviewer's comment for each point and then sketch how we plan to revise this manuscript to address these suggestions.

1. "Orbital forcing is not the major forcing of climate variability during the last millennium, except for the millennium-scale decreasing trend, is there any relationship between the upwelling and the volcano activities?"
We thank the reviewer for the suggestion. Yes, we actually meant the millennium timescales. We will examine the relationship between the upwelling and the volcano

activities, for instance by superposed epoch analysis. The volcanic forcing used in the simulations is available and it will be discussed in the revised manuscript.

2. "Since the link between the ISM and the upwelling is pronounced, how is the relationship between ENSO and the upwelling?"
We thank the reviewer for this thought. In the revised manuscript, we will also discuss the relationship between ENSO and the upwelling. However, it has to be borne in mind that the models still show some deficiencies in the simulation of ENSO events, and the analysis may yield different results for the models included in this study.

3. "The authors showed the CESM ensemble results, have you evaluated the ensemble spread to measure significance or uncertainties, for example including ensemble ranges would be useful."
We appreciate the reviewer's suggestion. However, maybe we have misunderstood the comment. The ensemble ranges can already be seen in our timeseries plots. Does the reviewer mean that we should consider all the ensemble members from the CESM last millennium ensemble?

4. "The future upwelling under RCP8.5 scenario reveals a negative trend, given that the upwelling favorable wind-stress is projected to increase. This is opposite from our common understanding that coastal upwelling at global scale would be intensified as the upwelling favorable wind-stress would be strengthened due to the enhanced air-sea temperature gradient under the global warming scenario. The authors explain this from the point of surface water stratification, is there a critical time when we could explain by the two different theories?"
We appreciate the reviewer for sharing this point. We will analyse the decadal trends in the future scenario to estimate when the water stratification overrides the strengthened wind-stress to affect the upwelling. This will be discussed in the revised manuscript.

**[CPD](CPD)**

Interactive
comment

---

## Author Comment (AC2) · 12 Aug 2019

We thank the reviewer for detailed reading of the manuscript and for the suggestions for improvement. In the following, we quote the reviewer's comment for each point and then sketch how we plan to revise this manuscript to address these suggestions.

1. "I would be very cautious about the choice of the models resolutions to be appropriate to investigate coastal upwelling in the Arabian Sea. The spatial extent (e.g., Figs. 1 a and b) shows that coastal areas are largely blank, especially for MPI-ESM, but also for CESM-CAM5. It is clear from previous studies that coastal upwelling is restricted to the vicinity of the coast, approx. 90 km offshore (Rixen et al., 2000). In fact, the

authors mention this issue on p.4 l.3 but miss to discuss what implications this might have on the reliability of the results. It was further discussed by Praveen et al. (2016 Geophys.Res.Lett.), that changes in upwelling under future warming scenarios are regionally limited and spatially inhomogeneous, finding that 1x1 degrees atmospheric resolution is not adequate for studying the coastal current system. The study of Yi et al. uses models with more than four times coarser resolutions and it has to be discussed if a potential increase of coastal vertical water transport might not be obscured by the coarse resolution."

We thank the reviewer for sharing this concern. We agree with the reviewer that the resolution might not be high enough to study the upwelling. However, as already mentioned in the manuscript, it is the best we can have so far. The models we chose have the highest resolution within the CMIP5 model pool which has been used by previous works to study upwelling systems (Wang et al., 2015 Nature). Our results, and the results of other previous studies that look into upwelling simulated in CMIP5 models, are conditioned on the limitations of these models. However, these results can still be useful to better interpret future studies with higher resolution models. In the revised manuscript, we will discuss in more detail how the resolution of the models affects the results.

2. "Although the approach of testing the models against observational data seems to be of minor importance in the study, I highly question the approach used here with G. bulloides data. My first concern is, that it is not clear from Fig. 2c) if cores RC2730 and RC2735 are actually within the modelled grid cell or outside. If the stations are covered, they are at the very edge of the models coverage anyway. Secondly, the actual correlation of the two time series, Oman margin upwelling intensity and G. bulloides abundance, is not given in Table 1, where I would expect it. It seems from the colour coding in Fig. 2 c) and d) that it is in the range of r=0.2 to 0.4. However, both models show a significant positive correlation only in the northern part of the basin, an area of open-ocean upwelling and outside the area that was used for calculating the time series. This is not explained in the text and I suppose that it is not appropriate to

verify the model results in this manner. I would also expect to see the G. bulloides data parallel to the upwelling time series data, as this would give a more obvious connection of the relationships. Especially as the authors discuss a "flip" at 1550 also be evident in the G. bulloides record, it has to be illustrated in a clearer way."

We appreciate the reviewer for these thoughts. We are aware that the location of the sediment records might not be optimal but we do not have many to choose from. The purpose of the comparison between the upwelling and the G.bulloides abundance is not to validate the model results against observation but to show the effect of external forcing over the last millennium. We also agree with the reviewer that the correlation should be shown more clearly. In the revised manuscript, we will also include a plot to illustrate the evolution of the G.bulloides record.

3. "The finding of increased upwelling favourable winds under RCP8.5 scenario together with a negative trend in upwelling intensity is explained with a likely overriding effect due to increased surface water stratification and warming SST. However, I miss a discussion of the results in context to previous studies of modelled upwelling in the Arabian Sea. Although using historical simulations over a much shorter period of time, Roxy et al. (2016 Geophys. Res. Lett.) for example found decreasing trends of phytoplankton productivity by using a similar set of simulations and also inferred an overriding effect of near-surface stratification as the main cause."

We thank the reviewer's suggestion. We will discuss the work of Roxy et al. (2016) in the revised manuscript. We would also like to point out that the validity of the study of Roxy et al. is also limited by the coarse spatial resolution of the models analysed in that study. Actually, that resolution is broadly similar to the spatial resolution of the models we have analysed in our manuscript and the models that were analysed in the Wang et al. manuscript cited in our previous response. Unfortunately, this is a limitation that can only be overcome when higher resolution simulation become available.